# Spinal Cord Epidural Stimulation Improves Lower Spine Sitting Posture Following Severe Cervical Spinal Cord Injury

**DOI:** 10.3390/bioengineering10091065

**Published:** 2023-09-09

**Authors:** Kundan Joshi, Enrico Rejc, Beatrice Ugiliweneza, Susan J. Harkema, Claudia A. Angeli

**Affiliations:** 1Kentucky Spinal Cord Injury Research Center, University of Louisville, Louisville, KY 40202, USA; kundan.joshi@louisville.edu (K.J.); enrico.rejc@louisville.edu (E.R.); beatrice.ugiliweneza@louisville.edu (B.U.); susan.harkema@louisville.edu (S.J.H.); 2Department of Bioengineering, University of Louisville, Louisville, KY 40292, USA; 3Department of Neurological Surgery, University of Louisville, Louisville, KY 40202, USA; 4Department of Medicine, University of Udine, 33100 Udine, Italy; 5Frazier Rehabilitation Institute, University of Louisville Health, Louisville, KY 40202, USA

**Keywords:** epidural stimulation, sitting posture, independence

## Abstract

Cervical spinal cord injury (SCI) leads to impaired trunk motor control, negatively impacting the performance of activities of daily living in the affected individuals. Improved trunk control with better sitting posture has been previously observed due to neuromuscular electrical stimulation and transcutaneous spinal stimulation, while improved postural stability has been observed with spinal cord epidural stimulation (scES). Hence, we studied how trunk-specific scES impacts sitting independence and posture. Fourteen individuals with chronic, severe cervical SCI with an implanted neurostimulator performed a 5-min tall-sit task without and with trunk-specific scES. Spine posture was assessed by placing markers on five spine levels and evaluating vertical spine inclination angles. Duration of trunk manual assistance was used to assess independence along with the number of independence changes and average independence score across those changes. With scES, the sacrum-L1 inclination and number of independence changes tended to decrease by 1.64 ± 3.16° (*p* = 0.07; Cohen’s d = 0.53) and 9.86 ± 16.8 (*p* = 0.047; Cohen’s d = 0.59), respectively. Additionally, for the participants who had poor sitting independence without scES, level of independence tended to increase by 12.91% [0%, 31.52%] (*p* = 0.38; Cohen’s d = 0.96) when scES was present. Hence, trunk-specific scES promoted improvements in lower spine posture and lower levels of trunk assistance.

## 1. Introduction

Spinal cord injury (SCI) is generally associated with impaired sensorimotor control below the level of injury. Consequently, injuries at the cervical level affect the ability to recruit hip and trunk musculature to maintain postural stability, which further limits upper extremity function during activities of daily living [1,2]. Sitting postural stability is also influenced by posture kinematics [3,4]. Upright posture in sitting is accomplished by activation of muscles to counteract the movement of the center of gravity of the trunk [3,5]. This ability is severely impacted by injuries above the C4 level, and it is also affected to varying degrees by injuries in the lower cervical and upper thoracic levels [6]. As a result, individuals with thoracic level SCI employ compensatory strategies such as using the upper limbs and non-postural muscles to maintain upright sitting posture [7]. If erect posture cannot be achieved, individuals adopt a more kyphotic posture that increases static sitting stability whilst also increasing the risk of low back conditions [1,6,8]. Better sitting posture is thus crucial to re-gaining trunk stability, which is one of the top rehabilitation goals of individuals with SCI [9].

Recovery of the ability to maintain sitting upright posture has been achieved to varying degrees with the reactivation of hip and trunk musculature via different electrical stimulation modalities [10,11,12,13,14,15]. Neuromuscular electrical stimulation (NMES) applied to the hip and trunk muscles of individuals with SCI led to improvements in pelvic tilt and shoulder height in static sitting [10,11]. Furthermore, it also led to the anterior shift of maximum interface pressure from the posterior sacral region of sacral sitters. This potentially reduced the risks of pressure ulcers for sacral sitters, while upright sitters demonstrated low interface pressure in the sacral region [12]. Closed-loop control systems have also been developed to maintain erect posture in static and dynamic sitting by stimulating abdominal muscles when the trunk is flexed over a pre-set threshold [1,16,17]. A combination of NMES with mat-based exercises also led to higher independence in reaching tasks [18]. In contrast to NMES that directly stimulates the agonist muscle responsible for the movement, transcutaneous spinal cord stimulation re-engages spinal neurocircuitry at different sites of the spinal cord that, in-turn, innervates the required muscles [13,14,15]. Consequently, stimulation of T11 and L1 regions in adults with SCI led to better static sitting posture characterized by decreased trunk angle and trunk curvature [14]. Stimulation at the same spinal levels in children with SCI also led to improved lower trunk angles in comparison to baseline relaxed sitting, and volitional effort to maintain erect posture without stimulation while postural control did not improve even with decreased lower trunk inclination induced by a passive anterior pelvic tilt [15]. Another neuromodulation modality, spinal cord epidural stimulation (scES), has also been used to re-engage spinal circuitry in individuals with motor complete SCI [19]. Previous studies have established that the lumbar spinal cord has the potential to evoke coordinated motor outputs similar to synergistic locomotor activity [20,21]. Optimization of scES parameters has led to the recovery of activation patterns for weight-bearing standing and locomotion [22,23]. However, the effects of scES on sitting posture or level of independence have not been studied. Increased trunk control and reach distances have been observed in individuals with thoracic SCI [24]. Decreased lumbar curvature was observed when scES was applied targeting the erector spinae and quadratus lumborum muscles during repeated trunk flexion/extension in reaching and retrieving tasks and during four-point kneeling movement [25]. Thus, scES has the potential to improve posture of individuals with SCI in static sitting which in turn could enable them to perform activities of daily living (ADLs) with increased independence.

Effects of epidural stimulation in stand, step, and reaching activities have been studied, while its effects on sitting posture and independence has not been explored [22,23,24]. This study aims to analyze the effects of trunk-specific scES on sitting posture and trunk control of individuals with chronic severe cervical SCI. We hypothesize that trunk-specific scES will improve spine curvature and level of independence compared to no stimulation.

## 2. Materials and Methods

### 2.1. Experimental Protocol

Fourteen individuals with chronic severe cervical SCI (Age: 37.8 ± 10.9 yrs.; time post-injury: 13.1 ± 10 yrs.; injury levels: C4–C6; American Spinal Injury Association Impairment Scale (AIS): A–C; body mass index (BMI) 16.7–27.9) were implanted with a scES unit (Table 1, Appendix A). Based on the BMI, eight participants were classified as normal, while three were classified as overweight and three as underweight. All the participants had sedentary lifestyles. Among them, eight participants used power wheelchairs; four used manual wheelchairs; and two used manual wheelchairs with SmartDrive. Caregivers were required on a 24-h basis for seven participants, and on a periodic basis for six participants, while one participant was independent. Seven participants required one assistant and a transfer board to perform transfers, while six participants required two assistants. The participant who did not require caregivers performed independent transfers using the transfer board. As previously reported, implantation consisted of a 16-electrode array implanted at the L1–S1 spinal cord level connected to an Intellis neurostimulator (Medtronic, Minneapolis, MN, USA) internalized in the lower back [26]. All participants provided written informed consent as described in the study protocol approved by the University of Louisville’s Institutional Review Board. The study was registered in clinical trials.gov prior to participant enrollment (NCT03364660). The participants had not undergone any training to improve sitting posture before enrollment. At the time of assessment, the participants were at different stages of a randomized controlled trial with 5 out of the 14 participants having received an intervention comprising trunk exercises. 

A full-body motion capture marker set (modified Helen Hayes model—Helen Hayes Hospital, West Haverstraw, NY, USA) and eight Kestrel motion capture cameras (Motion Analysis Corporation, Santa Rosa, CA, USA) were used to capture kinematics. Markers were placed on five spine levels—the C3/C4 marker was placed behind the neck, and the sacral marker was placed on the mid-point of the posterior superior iliac spines. The other three spine markers were placed on estimated locations on T10, T3/T4, and L1 spine levels such that the five markers divided the spine into four equidistant segments. Marker placements were carried out by experienced research staff. Participants were seated on a standard experiment mat to perform the designated activity.

Data were acquired at 100 Hz using Cortex software Version 6.2.3.1732 (Motion Analysis Corporation, Santa Rosa, CA, USA). The participants were instructed to sit as tall and stable as possible without upper limb support for 5 min. They were asked to call out the start of the activity which was initially performed without scES and then repeated with optimal trunk-specific scES. The stimulation configuration was optimized for each participant in order to enable sitting postural control. Due to the customization, configurations contained either a single cohort to facilitate maintenance of static sitting posture, as well as trunk extension and leaning, or multiple cohorts running simultaneously with specific functionalities (Appendix B). Since the implantation occurs at the L1–S1 spinal region that innervates primarily lower extremity muscle fibers, the electrode configurations are rostrally arranged such that the stimulation can be directed at the superior from the lumbar region towards the trunk and abdominal muscles.

Research staff provided assistance as needed to maintain safety or upright sitting during the assessments by supporting lower to midback to maintain posture (maximum assistance) or to prevent a backward fall (minimum to moderate assistance) or by supporting the sternum in case they were falling forward (Figure 1). The support was removed if the participant could regain sitting balance. Participants were also allowed to place their hands on the mat or their legs in case of loss of balance. A manual pulse was used to record changes in the level of external assistance provided to accomplish the task. 

### 2.2. Data Analysis

Marker identification in the kinematic data was performed in the Cortex software Version 6.2.3.1732 (Motion Analysis Corporation, Santa Rosa, CA, USA). In case of motion capture frames missing any marker data when blocked from view by research staff providing assistance, linear and polynomial interpolation techniques were used to remove missing gaps. Ortho Trak software Version 6.6.4 (Motion Analysis Corporation, Santa Rosa, CA, USA) was then used to generate 3D locations of the markers. A custom MATLAB code written in MATLAB Version R2017b (MathWorks, Natick, MA, USA) was used to isolate planar locations of the markers at the five spine levels (Sacrum, L1, T10, T3/T4, and C3/C4) in the sagittal plane (Figure 2a). Vertical inclination angles of the four spine segments were also calculated as below:(1)$$\theta_{S-L1}=\tan^{-1}(\frac{X_{L1}-X_{S}}{Z_{L1}-Z_{S}})$$
(2)$$\theta_{L1-T10}=\tan^{-1}(\frac{X_{T10}-X_{L1}}{Z_{T10}-Z_{L1}})$$
(3)$$\theta_{T10-T3T4}=\tan^{-1}(\frac{X_{T3/T4}-X_{T10}}{Z_{T3/T4}-Z_{T10}})$$
(4)$$\theta_{T3T4-C3C4}=\tan^{-1}(\frac{X_{C3/C4}-X_{T3/T4}}{Z_{C3/C4}-Z_{T3/T4}})$$
where $X_{marker}$ & $Z_{marker}$ are the horizontal and vertical coordinates of the specified marker, respectively; $\theta_{inferior\;marker-superior\;marker}$ is the inclination angle of the spine segment joining the specified inferior and superior spine markers. To quantify curvature of the spine, a least-squares circle was fitted onto the sagittal plane coordinates of the five markers, and a radius of the fit circle (radius of curvature) was obtained (Figure 2b). The method used to determine the radius of the fit circle was by the minimization of the equation of the circle after substituting the equation with the available five marker locations. Consider if the equation of the circle was:(5)$${\left(x-k\right)}^{2}+{\left(y-m\right)}^{2}=r^{2}$$
where (k, m) is the center of the circle, and r, the radius, the following equation expressed in the sagittal (XZ) plane was minimized to obtain best-fit values for k, m, and $r_{curvature}$:(6)$$F\left(k,\;m,\;r_{curvature}\right)=\sum [{\left(X_{marker}-k{)}^{2}+{\left(Z_{marker}-m\right)}^{2}-r_{curvature}{}^{2}\right]}^{2}$$
where $F\left(k,\;m,\;r_{curvature}\right)$ is the function to be minimized with respect to k, m, and $r_{curvature}$.

From the manual pulse data, events were separated where the participants were fully independent (labeled as “independent”). Likewise, the label “assisted” was assigned when they were externally supported by research staff due to loss of postural stability (“assisted trunk”) or if they placed their hands on their legs/the mat (“assisted hand/hands”). The duration of these events was summed up to obtain the fraction of the total 5 min duration where the participants were independent or needed to be assisted. The number of independence changes was also calculated from the events. For each event due to a change in independence, a custom independence score was used to rate the level of assistance needed. The ratings were from 0 (fully independent) to 6 (fully assisted on the trunk and both hands down). An average score was obtained by summing individual independence scores for all the events and dividing the sum by the number of events. 

### 2.3. Statistical Comparisons

Prior to statistical analysis, normality was evaluated using visual examination of variable distribution supplemented by evaluation of skewness and kurtosis. Acceptable normality was defined as having a distribution visually with normal and skewness between −2 and 2 and kurtosis between −7 and 7 [27]. Variables that satisfied these criteria were summarized with mean and standard deviation (SD) and analyzed by paired t-tests. Those that did not were summarized with interquartile range (25th and 75th percentiles) and analyzed by the signed rank test. The significance level was set to 5%. To quantify the observed changes, Cohen’s d effect sizes (ES = mean/SD) were used for normally distributed outcomes [28]. For non-normally distributed variables, a non-parametric equivalence to Cohen’s d was calculated as the median divided by the median absolute deviation (MAD), i.e., ES = median/MAD. Sawilowsky’s extension to Cohen’s criteria was used to classify the obtained effect sizes as very small (<0.2), small (0.2–0.49), median (0.5–0.79), large (0.8–1.19), very large (1.2–1.99), and huge (>2.0). The threshold of 0.5 was used for meaningfulness as it was found to be the minimally important difference in health-related quality of life [29,30].

## 3. Results

### 3.1. Posture Outcomes

There were no statistically significant differences observed for all four spine inclination angles as well as the radius of curvature when scES OFF and scES ON conditions were compared (Figure 3; Table 2). However, there was a medium-effect size change of 1.64 ± 3.16° (*p* = 0.07; Cohen’s d = 0.52) in the sacrum to the L1 inclination angle between scES OFF and ON conditions (Figure 3a). There was also a medium-effect-size increase of 1.9 ± 3.35° (*p* = 0.054; Cohen’s d = 0.57) in the L1 to T10 inclination angle when scES was applied (Figure 3b). There were no statistically significant changes observed in the anterior-posterior or vertical coordinate of all the spine markers between scES OFF and scES ON with all effect sizes being small or very small. Even though statistically significant differences were not found, the S.D. of the anterior-posterior and vertical coordinates of all the spine markers were lower in the scES ON condition compared to the scES OFF condition (Figure 3c).

### 3.2. Level of Independence

In the scES OFF condition, the participants performed the task independently for a median [IQR] duration of 98.5% [22.81%, 100%] of the total time and needed assisted for 1.5% [0%, 77.19%] (Table 2; Figure 4a). In the scES ON condition, they were independent for 100% [71.74%, 100%] and assisted for 0% [0%, 28.26%]. The 0.12% [0%, 12.65%] change in duration of independence (*p* = 0.18; Cohen’s d = 0.12) between the scES OFF and scES ON conditions was not statistically significant. However, there was a statistically significant mean decrease of −9.86 ± 16.8 in the number of independence changes (*p* = 0.047; Cohen’s d = 0.59). On the other hand, the −0.17 [−0.67, 0] change in median [IQR] independence score was not statistically significant (*p* = 0.34; Cohen’s d = 0.43) (Figure 4b,c). 

### 3.3. Subset Analysis

Based on the number of independence changes, a subset analysis was performed on N = 6 participants (A64, A97, A133, B40, B42, C193) by excluding the other 8 participants who showed a relatively better sitting postural control with scES OFF, as they were independent for more than 98% of the total time performing the task and had fewer than 4 independence changes in both scES OFF and scES ON conditions. There were no statistically significant changes observed in the spine inclination angles and the radius of curvature of the trunk (Table 3). However, compared to the scES OFF condition, there were medium-effect size changes of 2.25 ± 3.54° (*p* = 0.18; Cohen’s d = 0.63) in the sacrum to L1 inclination angle and of 1.62 ± 3.21° (*p* = 0.27; Cohen’s d = 0.5) in the T10 to T3/T4 inclination angle in the scES ON condition (Figure 5a,d). In addition, there was a large-effect size change of 3.2 ± 3.72° (*p* = 0.09; Cohen’s d = 0.86) in the L1 to T10 inclination angle (Figure 5b). There were no statistically significant changes observed in the anterior-posterior or vertical coordinate of all the spine markers between scES OFF and scES ON with effect sizes being small or very small. Interestingly, the S.D. of the anterior-posterior and vertical coordinates of all the spine markers were lower in the scES ON condition compared to the scES OFF condition (Figure 5c).

When comparing the level of independence and assistance, there were no statistically significant changes observed between the scES OFF and scES ON condition. However, a large-effect-size increase of 12.91% [0%, 31.52%] in median [IQR] duration of independence was observed (*p* = 0.38, Cohen’s d = 0.96). In addition, there was a statistically significant decrease of 22 ± 20.48 in the mean (SD) number of independence changes (*p* = 0.047, Cohen’s d = 1.07), along with a medium-effect-size change of 0.39 [−1.27, 0.08] in median [IQR] independence scores (*p* = 0.69, Cohen’s d = 0.57) from scES OFF to scES ON conditions. There was also a medium-effect-size change of 0% [−15.6%, 0%] in the median [IQR] duration of assistance with one hand on the mat (*p* = 0.5, Cohen’s d = 0.63) and a large-effect-size change of −4% [−8.87%, 0%] in duration of assistance on the trunk by research staff and one hand on the mat (*p* = 0.25, Cohen’s d = 1).

Among the participants in the subset analysis, one participant (C193) was assisted fully during the scES OFF condition and was fully independent during scES ON condition. During the 1st minute of the 5-min task, the assistance on the trunk enabled her to obtain upright posture in the scES OFF condition (Figure 6a), but her trunk tilted progressively anteriorly until the last minute (Figure 6c). In contrast, with scES, she started in an anteriorly tilted trunk posture to begin but corrected it progressively until she was fully upright especially in the lower back without needing any assistance (Figure 6b,c).

## 4. Discussion

The participants of this study were able to obtain more erect sitting posture in the lower spine with the aid of scES targeted at trunk stability compared to sitting without stimulation. The effects of scES on upper spine posture as well as overall curvature of the spine, however, were not statistically significant. In the presence of scES, the participants had decreased frequency in independence changes, indicating that stimulation enabled them to hold a certain independence level for longer periods. When a subset of participants was analyzed by excluding those participants with almost full independence and no more than four independence changes, sitting posture of the lower spine improved with stimulation along with obtaining a higher level of trunk independence, lower frequency of independence changes, and better independence scores. Results from one participant in the subset indicated that scES substituted the need of external trunk assistance throughout the 5-min task, and sitting posture progressed from an anterior trunk tilt in the first minute to an erect lower spine in the last minute. This target was unachievable for the participant without stimulation even with full assistance provided by research staff.

### 4.1. Posture and Independence with scES

Epidural stimulation optimized for trunk stability targets abdominal muscle groups responsible for trunk flexion and extension. This neuromodulation approach provides excitability to the muscles in the lower abdomen, and with the assistance of innervation from the residual circuitry, causes a resultant motor output [19,20,21]. For the tall-sit activity in our study, the excitation to the rectus abdominus and paraspinal muscles enabled the participants to modulate motor output and improve postural stability, independence, and maintain upright lower spine evidenced by a decrease in spine inclination of the Sacrum to L1 segment. The improved ability of the participants to hold an assistance or independence level with stimulation was reflected in the trend of the decreased spread of the anterior-posterior coordinates of spine markers from scES OFF to scES ON conditions. Analogous to these results, a previous study conducted with functional electrical stimulation of the trunk and hip extensors led to improved static shoulder height, pelvic tilt, and trunk lean in a fraction of participants [10]. Improved lower trunk extension, trunk curvature, and angle were observed along with increased activity in L3 level-erector spinae and oblique muscles as a result of transcutaneous stimulation of T11 and L1 spinal regions [13]. Upright posture of the L5 to S1, and pelvic to T8 segments, were observed due to transcutaneous stimulation of the T11 and L1 regions of children with SCI [15]. While achieving erect posture helps to maintain sitting stability, being able to perform the task without external assistance is also important. 

Not many studies have investigated the effects of stimulation on the level of independence; instead, the individuals with SCI were asked to perform certain tasks until they lost control or independence [11,13,24]. In one study, upon application of functional electrical stimulation on the quadriceps and gastrocnemius muscles, individuals with SCI were able to perform mat-based reaching and retrieving tasks with more independence scores in the Spinal Cord Independence Measure-III (SCIM-III) scale [18]. In our study, the median [IQR] level of independence was 98.5% [22.81%, 100%] without stimulation and 100% [71.74%, 100%] with stimulation. Because a fraction of participants (8/14) showed good independence without stimulation, the changes were statistically not significant. Hence, when we performed a subset analysis for the rest of the participants, there were improvements seen in the level of independence, number of independence changes, and independence score when scES was applied (Figure 5d–f, Table 3). The pattern of assistance provided also changed with decreases in the duration of one hand needing support with or without trunk support with stimulation (Table 3). It is likely that without stimulation, participants needed to put a hand on the mat to maintain stability more often than when stimulation was present. Even though there were no significant changes in spine marker coordinate locations in the presence of stimulation, the lower spine tended to be more erect (Figure 5c). Without stimulation, the lower spine tended to be more inclined even when more assistance was provided to aid in posture. This result reflected the strategy of the participants focusing on gaining independence and then on maintaining posture, a strategy that was highlighted in one participant (C193). In particular, without stimulation, she was fully assisted, but even with assistance, her posture progressively became anteriorly tilted. With stimulation, she did not need any external assistance, and her lower spine posture progressively became more erect from an initial anterior tilt (Figure 6a–c). These opposing responses indicate that epidural stimulation has the potential to substitute external assistance with an immediate effect. Continuous stimulation allows the individual to modulate neuromuscular activation resulting in spinal posture improvements along with improved independence. The transition from an anteriorly tilted to an erect posture also highlights the differences due to the intended duration of the task. Previous studies asked the participants to maintain static sitting bouts for as few as 10 s to a maximum of 1 min [13,15]. While smaller intended durations may maximize immediate improvements due to stimulation, the desirable effect would be steady improvements that last for a longer period. The ability to maintain better posture for a longer duration helps to perform other functional tasks that start from unsupported sitting such as trunk movements or reaching for objects [31]. Better sitting posture would also conceivably decrease the likelihood of pressure ulcers that arise from poor postures such as sacral sitting [6,8,12]. Given the participants’ sedentary lifestyles with around half of them requiring power wheelchairs, 24-h caretakers, and two persons to transfer, epidural stimulation has the potential to increase the range of potential achievable seated tasks that the participants could perform (Table 1). When taken together, these improved outcomes can help achieve better quality of life for individuals with SCI.

### 4.2. Physiological Response to scES

As previously reported, the epidural stimulator was implanted between L1 and S1 spinal cord levels, a section typically responsible for innervating hip muscle fibers [19,20,21,22,23]. On the other hand, nerve and muscle fibers responsible for trunk functions are present in the thoracic spinal cord. Consequently, trunk specific scES configurations are generally rostral with cathodes lining up the first row of the implanted electrode array and allowing for tonic stimulation to be projected from the lumbar region towards the pelvis and the trunk muscles (Appendix B). Tonic stimulation in combination with residual descending supraspinal drive across the lesion enables the participant to maintain static balance, independence, and posture [19,23,24]. This approach leads to a focal pattern of motor unit recruitment to maintain posture, analogous to the autonomous regulation of posture in uninjured population [24,32]. This coordinated activation pattern as a result of L1 level epidural stimulation was most likely responsible for improving lower spine posture, and the ability to hold a level of independence/assistance, as well as drastically improving the level of independence for the participants who did not have a high level of pre-existing trunk independence without stimulation.

The potential of epidural stimulation to improve seated posture and independence had not been investigated previously. Our results highlight the need for further research on neurophysiological changes leading to changes in seated posture in various functional tasks brought about by epidural stimulation, in a manner similar to generation of locomotor patterns with stimulation of the lumbosacral spinal cord [20,21,22]. Comparatively more pronounced improvements in posture were observed in past studies that used transcutaneous spinal cord stimulation for neuromodulation [13]. However, the increased ease of using the trunk-specific scES configuration in home and community environments implies the need for further studies to link stimulation with posture improvements in seated tasks, as well as improvements in execution of activities of daily living. All the participants in the study had sedentary lifestyles with 5 out of 14 participants having undergone an intervention program comprising trunk exercises at the time of assessment of spine posture (Appendix A). The improvements seen in lower spine posture regardless of different stages of intervention suggests that trunk-specific scES could be a beneficial tool for them to increase their level of activity overall and potentially increase independence during activities of daily living (Table 1).

### 4.3. Limitations

There was a lack of validated assessments of sitting posture as well as limited past literature. Hence, comparative analysis of the results of our study with past studies was not intuitive. The participants’ characteristics in the current study ranged in age, gender, level of injury, time since injury, and AIS scores and sitting ability prior to implantation. All these add to the variability in our sample population.

## 5. Conclusions

The effect-size analysis of sitting posture suggested that epidural stimulation targeted at gaining trunk stability led to improved upright lower spine control in the individuals with SCI enrolled in the present study. Given the variations in participant demographics, their injury levels, and initial trunk control, the primary hypothesis of improvements in overall trunk curvature was not achieved. The stimulation did lead to improved independence in performing the static sitting for individuals who did not already have good independence without stimulation, while also significantly reducing the frequency of changes in independence levels of all participants to maintain balance and posture. These results validate our second hypothesis that trunk-specific stimulation leads to improved independence in those with more severe motor control impairments. Further research should focus on standardization of the assessment of sitting posture. Future work can also entail how stimulation substitutes external assistance needed to maintain posture and if/how it stabilizes with time. The long-term effects of stimulation can also be manifested in rehabilitation programs aimed at improving trunk control and posture, which would in turn lead to better quality of life for the individuals with SCI.

## Figures and Tables

**Figure 1 bioengineering-10-01065-f001:**
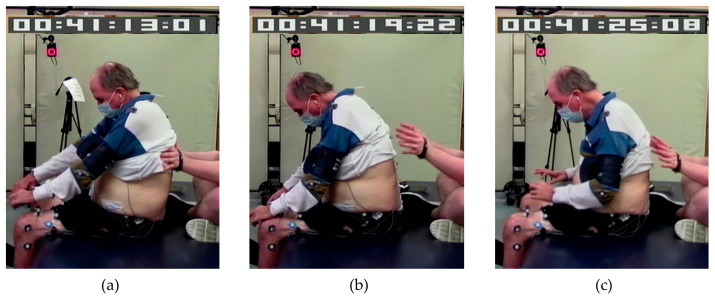
A participant requiring various levels of assistance to complete the task: (**a**) posterior trunk assistance from research staff, (**b**) anterior assistance by placing hands on legs, and (**c**) no external assistance.

**Figure 2 bioengineering-10-01065-f002:**
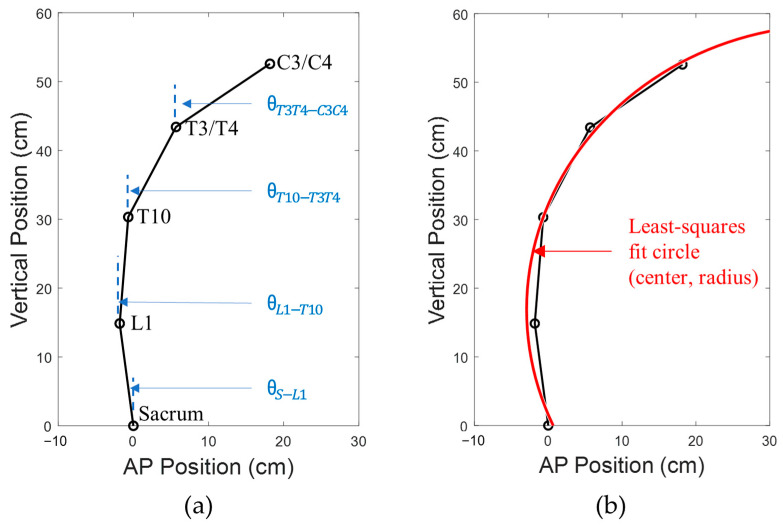
Calculation of (**a**) vertical spine inclination angles for the spine segments, and (**b**) least-squares fit circle to obtain radius of curvature.

**Figure 3 bioengineering-10-01065-f003:**
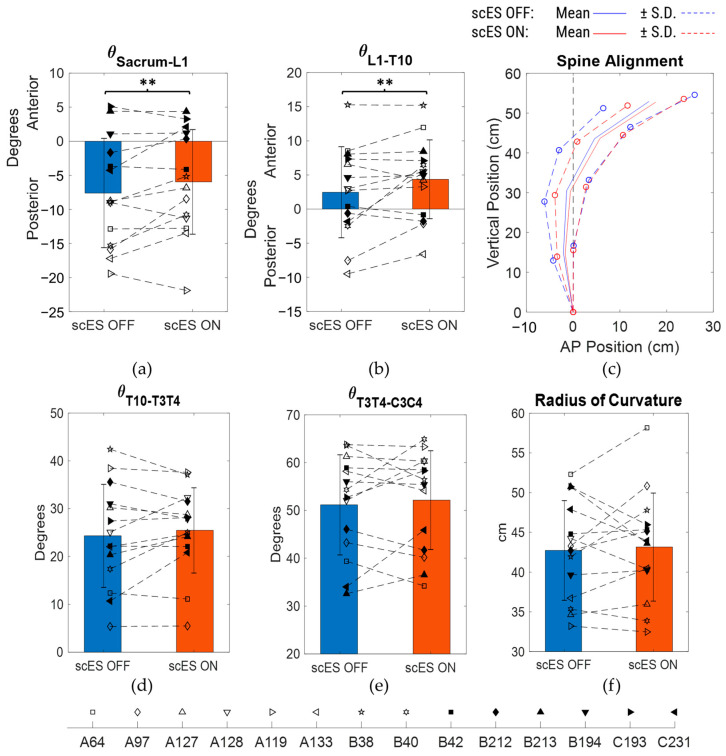
Comparison of spine inclination angles, spine posture, and radius of curvature between scES OFF and scES ON conditions: (**a**) sacrum to L1 angle, (**b**) L1 to T10 angle, (**c**) mean ± S.D. of spine marker coordinates, with sacrum as reference origin, (**d**) T10 to T3/T4 angle, (**e**) T3/T4 to C3/C4 angle, and (**f**) radius of curvature obtained from least-squares fit circle. Note: ** implies *p*-value not significant but Cohen’s d ≥ 0.5.

**Figure 4 bioengineering-10-01065-f004:**
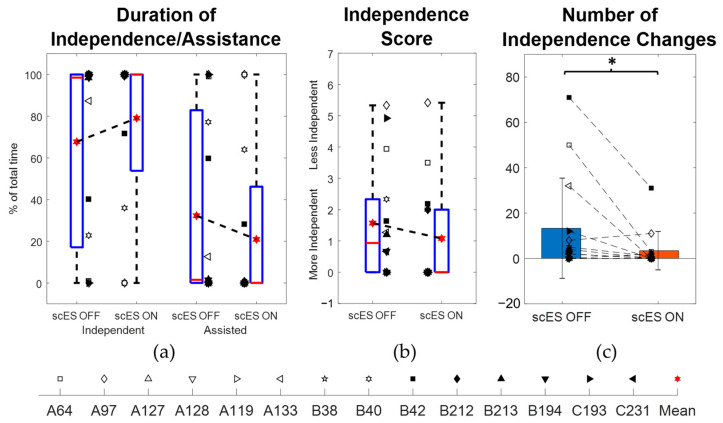
Comparison of independence outcomes between scES OFF and scES ON conditions: (**a**) percentage duration of independence and needing assistance, (**b**) independence score, and (**c**) number of independence changes. Note: * implies *p*-value < 0.05.

**Figure 5 bioengineering-10-01065-f005:**
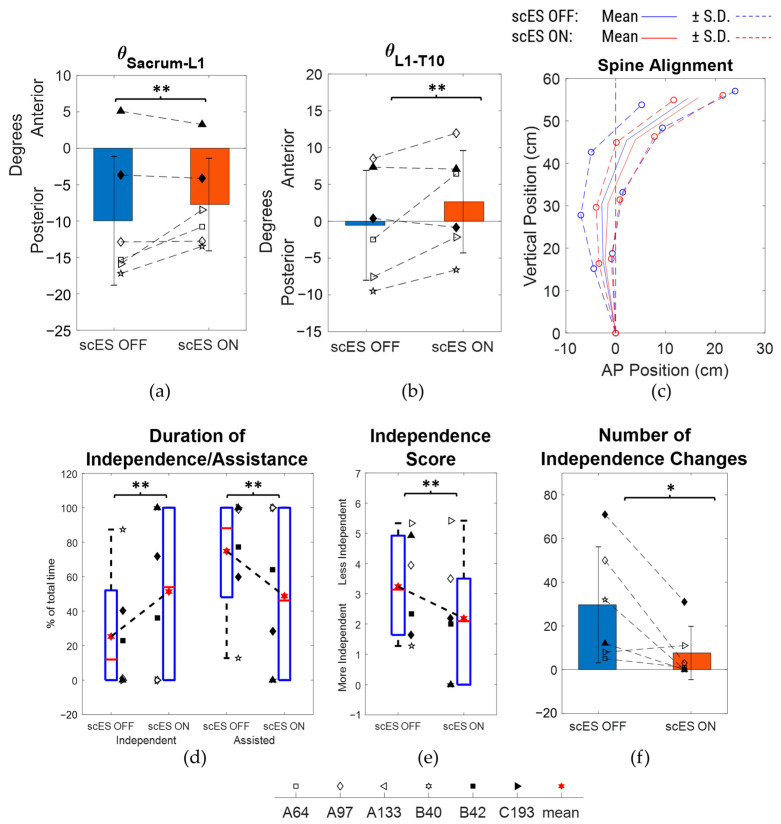
Comparison of spine inclination angles, spine posture, and radius of curvature between scES OFF and scES ON conditions for the subset (N = 6): (**a**) sacrum to L1 angle, (**b**) L1 to T10 angle, (**c**) mean ± S.D. of spine marker coordinates, with sacrum as reference origin, (**d**) T10 to T3/T4 angle, (**e**) T3/T4 to C3/C4 angle, and (**f**) radius of curvature obtained from least-squares fit circle. Note: * implies *p*-value < 0.05; ** implies *p*-value not significant but Cohen’s d ≥ 0.5.

**Figure 6 bioengineering-10-01065-f006:**
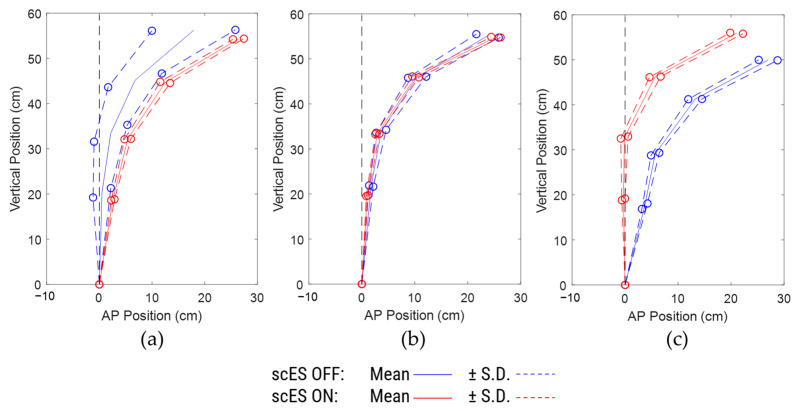
Comparison of spine posture for different 1-min periods for participant C193: (**a**) 1st minute, (**b**) 3rd minute, and (**c**) 5th minute.

**Table 1 bioengineering-10-01065-t001:** Participant Demographics.

ID	Gender	Age	Time Since Injury	Neuro Level	AIS	BMI	Mobility	Lifestyle	Caregiver	Transfers
A64	M	55.7	38.6	C4	A	21.6	Power wheelchair	Sedentary	24 h	Two-person
A97	M	38.6	15.4	C4	A	16.7	Power wheelchair	Sedentary	24 h	Two-person
A127	F	40.6	13.1	C4	A	23.9	Manual wheelchair	Sedentary	Periodic	One person Transfer board
A128	M	38.6	8.7	C4	A	23.7	Manual wheelchair	Sedentary	Periodic	One person Transfer board
A119	F	26.4	11.5	C5	A	17.6	Power wheelchair	Sedentary	Periodic	Two-person
A133	M	34.6	13.1	C5	A	27.9	Power wheelchair	Sedentary	Periodic	One person Transfer board
B38	M	21.9	3.3	C4	B	26.8	Power wheelchair	Sedentary	Periodic	Two-person
B40	M	36.0	4.7	C4	B	20.3	Power wheelchair	Sedentary	24 h	Two-person
B42	M	58.7	7.5	C4	B	26.8	Power wheelchair	Sedentary	24 h	Two-person
B212	F	47.9	29.7	C4	B	24.8	Manual wheelchair	Sedentary	Periodic	One person Transfer board
B213	F	25.1	5.4	C4	B	19.5	Power wheelchair	Sedentary	24 h	One person Transfer board
B194	F	37.8	18.2	C5	B	17.9	Manual wheelchair + SmartDrive	Sedentary	Independent	Independent Transfer board
C193	F	41.1	8.9	C4	C	24.4	Manual wheelchair + SmartDrive	Sedentary	Periodic	One person Transfer board
C231	F	26.8	5.7	C6	C	23.5	Manual wheelchair	Sedentary	24 h	One person Transfer board

Note: AIS—American Spinal Injury Association Impairment Scale, BMI—Body Mass Index.

**Table 2 bioengineering-10-01065-t002:** Inferential statistics for posture and independence outcomes.

Parameter	scES OFF	scES ON	Change	*p*-Value	Effect Size	Classification
θ*_S-L1_* (°)	−7.6 ± 8.02	−5.96 ± 7.68	1.64 ± 3.16	0.07	0.52 **	Medium
θ*_L1-T10_* (°)	2.46 ± 6.67	4.36 ± 5.78	1.9 ± 3.35	0.054	0.57 **	Medium
θ*_T10-T3T4_* (°)	24.32 ± 10.76	25.46 ± 8.9	1.14 ± 4.6	0.37	0.25	Small
θ*_T3T4-C3C4_* (°)	51.16 ± 10.48	52.15 ± 10.34	1 ± 6.06	0.55	0.16	Very small
r_curvature_ (cm)	42.72 ± 6.26	43.15 ± 6.79	0.42 ± 4.41	0.72	0.1	Very small
% Independent	98.5 [22.81, 100]	100 [71.74, 100]	0.12 [0, 12.65]	0.18	0.12	Small
% Assisted	1.5 [0, 77.19]	0 [0, 28.26]	−0.12 [−12.65, 0]	0.18	0.12	Small
No. of Independence Changes	13.29 ± 22.1	3.43 ± 8.46	−9.86 ± 16.8	**0.047 ***	0.59 **	Medium
Independence Score	0.93 [0, 2.33]	0 [0, 2]	−0.17 [−0.67, 0]	0.34	0.43	Small

Note: Statistics are mentioned in the formats—mean ± SD or median [IQR] depending on the validity of assumption of normality. * implies *p*-value < 0.05. ** implies Cohen’s d ≥ 0.5.

**Table 3 bioengineering-10-01065-t003:** Inferential statistics of the subset (N = 6) for posture and independence outcomes.

Parameter	scES OFF	scES ON	Change	*p*-Value	Effect Size	Classification
θ*_S-L1_* (°)	−9.97 ± 8.84	−7.73 ± 6.36	2.25 ± 3.54	0.18	0.63 **	Medium
θ*_L1-T10_* (°)	−0.55 ± 7.46	2.65 ± 6.95	3.2 ± 3.72	0.09	0.86 ******	Large
θ*_T10-T3T4_* (°)	17.78 ± 7.92	19.4 ± 8.99	1.62 ± 3.21	0.27	0.5 **	Medium
θ*_T3T4-C3C4_* (°)	51.13 ± 8.05	51.7 ± 11.9	0.58 ± 6.21	0.83	0.09	Very small
r_curvature_ (cm)	43.86 ± 6.97	45.77 ± 8.36	1.9 ± 4.67	0.37	0.41	Small
% Independent	11.88 [0, 40.23]	53.87 [0, 100]	12.91 [0, 31.52]	0.38	0.96 **	Large
% Assisted	88.12 [59.77, 100]	46.13 [0, 100]	−12.91 [−31.52, 0]	0.38	0.96 **	Large
% Assisted (1 Hand Only)	0 [0, 15.6]	0 [0, 0]	0 [−15.6, 0]	0.5	0.63 **	Medium
% Assisted (2 Hands)	5.94 [0, 17.44]	0 [0, 5.97]	0 [−11.47, 0]	1	0.23	Small
% Assisted (Trunk Only)	22.89 [1.33, 53.59]	1.18 [0, 20.58]	−1.21 [−21.75, 0]	0.38	0.19	Very Small
% Assisted (Trunk + 1 Hand)	4 [0, 14.73]	0 [0, 0]	−4 [−8.87, 0]	0.25	1 **	Large
% Assisted (Trunk + 2 Hands)	8.87 [0, 31.4]	0.61 [0, 1.7]	−0.38 [−15.75, 1.7]	1	0.06	Very Small
No. of Independence Changes	29.67 ± 26.52	7.67 ± 12.16	−22 ± 20.48	**0.047 ***	1.07 **	Large
Independence Score	3.14 [1.64, 4.92]	2.09 [0, 3.5]	−0.39 [−1.27, 0.08]	0.69	0.57 **	Medium

Note: Statistics are mentioned in the formats—mean ± SD or median [IQR] depending on the validity of assumption of normality. * implies *p*-value < 0.05. ** implies Cohen’s d ≥ 0.5.

## Data Availability

The data presented in this study are available on request from the corresponding author. The data are not publicly available due to lack of available repository.

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
