# Peer review of "Spinal Cord Epidural Stimulation Improves Lower Spine Sitting Posture Following Severe Cervical Spinal Cord Injury"

_bioengineering, 2023, doi:10.3390/bioengineering10091065_

Round 1
Reviewer 1 Report
The purpose of this paper was to investigate the effects of spinal epidural stimulation on sitting posture and trunk control function in patients with chronic severe cervical spinal cord injury. However, the results presented in the article show that the effects of spinal cord epidural stimulation on upper spine posture and overall curvature of the spine was not statistically significant. These results are insufficient to support the article title and conclusions.
Author Response
Response to Reviewer #1:
Comment #1:
The purpose of this paper was to investigate the effects of spinal epidural stimulation on sitting posture and trunk control function in patients with chronic severe cervical spinal cord injury. However, the results presented in the article show that the effects of spinal cord epidural stimulation on upper spine posture and overall curvature of the spine were not statistically significant. These results are insufficient to support the article title and conclusions.
Response:
As pointed out by the reviewer, we have changed the title of the study to reflect the improvements in lower spine posture. However, in the conclusions, we have only stated that stimulation improves lower spine posture which are supported by the results that were seen in the study. Hence, no changes were made in the conclusion.
Changed Line #2-3 (Title):
Spinal cord epidural stimulation improves lower spine sitting posture following severe cervical spinal cord injury.

Reviewer 2 Report
This manuscript examined the influence of trunk-specific scES (spinal cord electrical stimulation) on the sitting posture and trunk control of fourteen people with chronic, severe cervical SCI (spinal cord injury). Trunk-specific scES promoted improvements in lower spine posture and lower levels of trunk assistance. The manuscript is highly original, the data is reliable, and the reasoning is sound. The study was done carefully and the supporting data were elaborated. Following the minor revisions, I am delighted to be able to publish this captivating paper.
1. A few additional details that shine light on what is included in the introduction would benefit the paper and improve sentence flow. However, lots of studies and reviews were written on re-engaging spinal circuitry in individuals with motor complete SCI using scES. The introduction reference should be renewed.
2. The methods used are clear and replicable. All the result appear to have been generated using the methodology described. More detailed information about optimal trunk-specific scES is needed.
3. More discussion of the clinical and research implications of the findings in the study should be added
Author Response
Response to Reviewer 2:
Comments:
This manuscript examined the influence of trunk-specific scES (spinal cord electrical stimulation) on the sitting posture and trunk control of fourteen people with chronic, severe cervical SCI (spinal cord injury). Trunk-specific scES promoted improvements in lower spine posture and lower levels of trunk assistance. The manuscript is highly original, the data is reliable, and the reasoning is sound. The study was done carefully and the supporting data were elaborated. Following the minor revisions, I am delighted to be able to publish this captivating paper.
Comment #1:
A few additional details that shine light on what is included in the introduction would benefit the paper and improve sentence flow. However, lots of studies and reviews were written on re-engaging spinal circuitry in individuals with motor complete SCI using scES. The introduction reference should be renewed.
Response:
To improve sentence flow, we added the following sentences: (Lines 66-68) as well as added references 20 and 21.
Previous studies have established that the lumbar spinal cord has the potential to evoke coordinated motor outputs similar tosynergistic locomotor activity [20, 21].
Comment #2.
The methods used are clear and replicable. All the result appear to have been generated using the methodology described. More detailed information about optimal trunk-specific scES is needed.
Response:
Information about the optimal trunk-specific scES was previously presented in Appendix A1 of the original submission. However, as per the reviewer’s suggestions, information regarding trunk-specific scES has been added to the Methods section with a reference to the appendix section that contains detailed configurations.
Added Lines 114-120:
The stimulation configuration was optimized for each participant in order to enable sitting postural control. Due to the customization, configurations contained either a single cohort to facilitate maintenance of static sitting posture, as well as trunk extension and leaning, or multiple cohorts running simultaneously with specific functionalities (Appendix A1). Since the implantation occurs at the L1-S1 spinal region that innervates primarily lower extremity muscle fibers, the electrode configurations are rostrally arranged such that the stimulation can be directed superior from the lumbar region towards the trunk and abdominal muscles.
Comment #3:
More discussion of the clinical and research implications of the findings in the study should be added.
Response:
As requested by the reviewer, the clinical and research implications of the findings were added to the discussion section. In conjunction with Reviewer #3’s comment #1, information about the lifestyles of the participants were added, and how obtaining postural improvements would help them achieve their lifestyle goals were described. Research implications in terms of how the study addresses a gap in current literature were updated.
The lines 335-337 and lines 356-367 were added:
Given their sedentary lifestyles with around half of them requiring power wheelchairs, 24-hr caretakers, and two persons to transfer, epidural stimulation has the potential to increase the range of potential achievable seated tasks that the participants could perform (Table 1).
The potential of epidural stimulation to improve seated posture and independence had not been investigated previously. Our results highlight the need for further research on neurophysiological changes leading to changes in seated posture in various functional tasks brought about by epidural stimulation, in a manner similar togeneration of locomotor patterns with stimulation of the lumbosacral spinal cord [20-22]. Comparatively more pronounced improvements in posture were observed in past studies that usedtranscutaneous spinal cord stimulation for neuromodulation [13]. However, the increased ease of using the trunk-specific scES configuration in home and community environments implies the need for further studies to link stimulation with posture improvements in seated tasks, as well as improvements in execution of activities of daily living. Given that all the participants in the study had sedentary lifestyles, trunk-specific scES could be a beneficial tool for them to increase their level of activity overall and potentially increase independence during activities of daily living (Table 1).

Reviewer 3 Report
After analyzing the article, the following concerns arose:
1) In the description of the people who participated in the study, there is no information about their lifestyle, physical activity, and possible information about injuries in the studied spine-related musculoskeletal system. This is crucial information, and such an interview and analysis is the basis of the research presented (I encourage you to study publications such as DOI:10.3390/s23115004).
2) The description of the measurement technology is at a low level and does not allow the authors to decode the study.
3) The article's description lacks research, e.g., with the coaxing of EMG technology and measurements of muscle tension in the muscular system around the spine.
4) Medical information describing the medical/physiological condition of the spines in the subjects is missing from the description of the issues. It would be interesting to combine this information with the description from the medical side.
5) The authors had not included a description of the originality of the research conducted. This should be significantly modified by highlighting the authorial/original elements.
Author Response
Response to Reviewer #3:
Comment:
After analyzing the article, the following concerns arose:
Comment #1:
In the description of the people who participated in the study, there is no information about their lifestyle, physical activity, and possible information about injuries in the studied spine-related musculoskeletal system. This is crucial information, and such an interview and analysis is the basis of the research presented (I encourage you to study publications such as DOI:10.3390/s23115004).
Response:
As pointed out by the reviewer, we updated the information about the lifestyle of the participants as well as wheelchair usage, need for caregivers, and the ability to transfer. This information was added to Table 1. Regarding information about the injuries, they were previously addressed in Table 1 of the original submission. All of the injuries were cervical level injuries with majority of the injuries being motor complete (AIS A and B), and 2 participants were motor incomplete (AIS C). We added a brief description of the participant demographics in the Methods sections and referenced the updated Table 1.
Lines 88-93 were added:
All the participants had sedentary lifestyles. Among them, eight participants used power wheelchairs, 4 used manual wheelchairs and 2 used manual wheelchairs with SmartDrive. Caregivers were required on a 24-hr basis for 7 participants, and on a periodic basis for 6 participants, while one participant was independent. Seven participants required one assistant and a transfer board to perform transfers, while six participants required two assistants. The participant who did not require caregivers performed independent transfers using the transfer board.
Comment #2:
The description of the measurement technology is at a low level and does not allow the authors to decode the study.
Response:
As requested by the reviewer, further description of the setup of motion capture, as well as post-processing steps were added. These details should make it easier for readers to decode/replicate the protocol of the study.
Lines 103-109 were added & Lines 134-136 were added:
Markers were placed on 5 spine levels – C3/C4 marker was placed behind the neck and sacral marker was placed on the mid-point of the posterior superior iliac spines. The other three spine markers were placed on estimated locations on T10, T3/T4, and L1 spine levels such that the five markers divided the spine into 4 equidistant segments. Marker placements were carried out by experienced research staff. Participants were seated on a standard experiment mat to perform the designated activity.
In case of motion capture frames missing any marker data when blocked from view by research staff providing assistance, linear and polynomial interpolation techniques were used to remove missing gaps.
Comment #3:
The article's description lacks research, e.g., with the coaxing of EMG technology and measurements of muscle tension in the muscular system around the spine.
Response:
During the experiments, we did acquire electromyography of the trunk, abdomen and leg muscles for various sitting postural control activities in addition to tall-sit. However, when stimulation is present, a lot of stimulation artifacts are observed in the EMG of the muscles. In our opinion, providing methods for filtering stimulation artifacts from EMG data and the analysis of the neuromuscular changes associated with tall sit and other postural activities performed with and without stimulation would constitute a whole separate study, and hence, would not be in the purview of this submission.
Comment #4:
Medical information describing the medical/physiological condition of the spines in the subjects is missing from the description of the issues. It would be interesting to combine this information with the description from the medical side.
Response:
Similar to the previous comment, the analysis of the medical/physiological condition of the spine would constitute a separate paper itself. Moreover, there has not been research on correlating the information to sitting posture. While we agree with the reviewer that it could provide interesting insights, we believe adding such an analysis would not impact the results / interpretation of our study.
Comment #5:
The authors had not included a description of the originality of the research conducted. This should be significantly modified by highlighting the authorial/original elements.
Response:
As suggested by the reviewer, at the end of introduction, the reason for the research and how it addresses gaps in literature are added.
Added lines 78-79:
Effects of epidural stimulation in stand, step, and reaching activities have been studied, while its effects on sitting posture and independence has not been explored.

Round 2
Reviewer 3 Report
After analyzing the revised article, it should be noted that the following questions and concerns have arisen, which will give new quality and enrich the descriptions presented:
1) What were the characteristics of the individuals in view of BMI and their physical activity. This gives a lot of additional information to supplement the characteristics of the subjects (see, for example, https://doi.org/10.3390/s23115004).
2) There is no information on whether the subjects already had injuries in the area of the intervertebral disc or appeared, for example, osteophytes.
3) There is no information on whether the subjects underwent a set of exercises under the guidance of a physical therapist before the study, who prepared them for this type of study through a series of exercises.
The publication, after completing the above content, will be able to undergo further publishing.
Author Response
Response to Reviewer #3:
Comment:
After analyzing the revised article, it should be noted that the following questions and concerns have arisen, which will give new quality and enrich the descriptions presented:
Comment #1:
What were the characteristics of the individuals in view of BMI and their physical activity. This gives a lot of additional information to supplement the characteristics of the subjects (see, for example, https://doi.org/10.3390/s23115004).
Response:
As per the reviewer’s suggestions, details about BMI were included in the Methods section and Table 1.
Lines 86-90 were edited as follows:
Fourteen individuals with chronic severe cervical SCI (Age: 37.8 ± 10.9 yrs.; time post-injury: 13.1 ± 10 yrs.; injury levels: C4-C6; American Spinal Injury Association Impairment Scale (AIS): A-C; body mass index (BMI) 16.7–27.9) were implanted with a scES unit (Table 1, Appendix A). Based on the BMI, 8 participants were classified as normal, while 3 as overweight and 3 as underweight.
Comment #2:
There is no information on whether the subjects already had injuries in the area of the intervertebral disc or appeared, for example, osteophytes.
Response:
As per the reviewer’s suggestion, information about the spine obtained from MRI reports prior to implantation surgery has been included in Appendix A, Table A1. Since only a few participants showed mildly degenerative conditions in the spine such as mild desiccation of intervertebral discs or osteophyte presentation, the information was not included in the main manuscript.
To accommodate the addition of Appendix A, the previous Appendix A was changed to Appendix B.
Comment #3:
There is no information on whether the subjects underwent a set of exercises under the guidance of a physical therapist before the study, who prepared them for this type of study through a series of exercises.
Response:
Individuals reported in this study were participating in a randomized trial with multiple interventions. As per the study protocol, the participants did not receive standardized therapy prior to enrollment in the trial. The main objective of this study was to report on the comparison of stimulation to no stimulation at the assessment time point. Due to the low number of participants in the different intervention points, a comparison of groups was not substantiated. Nine individuals had not received any training related to sitting posture at the assessment time point, while five had received an intervention where trunk exercises were a component of the training. This study emphasizes that the changes brought about by stimulation regardless of starting ability.
Hence as per the reviewer’s suggestion, Lines 102-105 were added in the Methods section and Lines 372-378 were edited in the Discussion section. Information regarding those participants who performed trunk-specific intervention was also included in Table A1 of Appendix A.
The participants had not undergone any training to improve sitting posture before enrollment. At the time of assessment, the participants were at different stages of a randomized controlled trial with 5 out of the 14 participants having received an intervention comprising trunk exercises.
All the participants in the study had sedentary lifestyles with 5 out of 14 participants having undergone an intervention program comprising trunk exercises at the time of assessment of spine posture (Appendix A). The improvements seen in lower spine posture regardless of different stages of intervention suggests that trunk-specific scES could be a beneficial tool for them to increase their level of activity overall and potentially increase independence during activities of daily living (Table 1).

Round 3
Reviewer 3 Report
The article in its current form may be subject to further publishing process. I make no more comments.